# All-Tunicate Cellulose Film with Good Light Management Properties for High-Efficiency Organic Solar Cells

**DOI:** 10.3390/nano13071221

**Published:** 2023-03-29

**Authors:** Chen Jiang, Meiyan Wu, Fang Zhang, Chao Liu, Mingliang Sun, Bin Li

**Affiliations:** 1School of Materials Science and Engineering, Ocean University of China, Qingdao 266100, China; 2CAS Key Laboratory of Biofuels, Qingdao Institute of Bioenergy and Bioprocess Technology, Chinese Academy of Sciences, Qingdao 266101, China; wumy@qibebt.ac.cn (M.W.); liuchao@qibebt.ac.cn (C.L.); 3National Engineering Research Center for Nanotechnology, Shanghai 200241, China; fangzhang@alumni.sjtu.edu.cn; 4Lignocellulose Biorefinery Laboratory, Shandong Energy Institute, Qingdao 266101, China; 5Metabolomics Group, Qingdao New Energy Shandong Laboratory, Qingdao 266101, China

**Keywords:** cellulose composite films, cellulose nanofibrils, micro-fibrillated cellulose, tunicate, organic solar cells

## Abstract

Tunicate nanocellulose with its unique properties, such as excellent mechanical strength, high crystallinity, and good biodegradability, has potential to be used for the preparation of light management film with tunable transmittance and haze. Herein, we prepared a whole tunicate cellulose film with tunable haze levels, by mixing tunicate microfibrillated cellulose (MFC) and tunicate cellulose nanofibrils (CNF). Then, the obtained whole tunicate cellulose film with updated light management was used to modify the organic solar cell (OSC) substrate, aiming to improve the light utilization efficiency of OSC. Results showed that the dosage of MFC based on the weight of CNF was an important factor to adjust the haze and light transmittance of the prepared cellulose film. When the dosage of MFC was 3 wt.%, the haze of the obtained film increased 74.2% compared to the pure CNF film (39.2%). Moreover, the optimized tunicate cellulose film exhibited excellent mechanical properties (e.g., tensile strength of 168 MPa, toughness of 5.7 MJ/m^3^) and high thermal stability, which will be beneficial to the workability and durability of OSC. More interestingly, we applied the obtained whole tunicate cellulose film with a high haze (68.3%) and high light transmittance (85.0%) as an additional layer to be adhered to the glass substrate of OSC, and a notable improvement (6.5%) of the power conversion efficiency was achieved. With the use of biodegradable tunicate cellulose, this work provides a simple strategy to enhance light management of the transparent substrate of OSC for improving power conversion efficiency.

## 1. Introduction

Light haze is defined as the percentage of the transmitted light intensity that deviates more than 2.5° from the angle of the incident light to the total transmitted light intensity [1,2], and it is an important parameter to evaluate a light management substrate for capturing light. Ideally, a light management substrate used in solar cells and organic light-emitting diode (OLED) devices should be a transparent substrate with both high haze and high light transmittance, but low light reflection, in order to capture and utilize light as much as possible [3,4]. A high haze of a high transparent substrate can increase the forward scattering of light and decrease light reflection [5], which can effectively improve the ability of solar cells and OLED devices to capture light, leading to a high photoelectric conversion efficiency [6,7,8,9]. Yet, conventional transparent substrates with high light transmittance, such as glass and plastics, usually exhibit low haze [10]. Hence, they are not ideal transparent substrates for solar cells. Recently, a variety of techniques have been developed to treat transparent substrates to achieve a high haze. For instance, by introducing nanomaterials or using mechanical methods to obtain a rough surface, the haze of glass can be significantly enhanced [11,12,13,14]. The haze of plastic films can be improved by introducing air gaps or nanostructures [15,16,17]. However, these processes are relatively complex and expensive. In addition, other disadvantages of glass substrates, such as fragility, relatively high weight, and plastic substrates, e.g., high thermal expansion coefficient, low thermal stability, and non-biodegradability, limit their practical applications [18]. Therefore, it is highly needed to develop a renewable, degradable, and easy-to-prepare light management substrate with both high light transmittance and high haze.

Cellulose is the most abundant natural polymer material on earth, and it exists in plant cell walls, algae, and some tunicate animals. In recent years, cellulose films have been used as light management substrates in solar cells [8,19,20,21,22] and OLED [5,23,24], due to their advantages of renewability, biodegradability, good thermal stability, low density and good flexibility. For instance, a cellulose film with a light transmittance of 96% and a haze of 60% was prepared from 2,2,6,6-tetramethylpiperidine-1-oxyl (TEMPO) oxidized wood fibers by a simple vacuum filtration process. The obtained cellulose film was used as the transparent substrate of organic solar cells (OSC), and it was found that 10% improvement of power conversion efficiency (PCE) was achieved [25]. Except for cellulose fibers, wood-based microfibrillated cellulose (MFC) and cellulose nanofibrils (CNF) with smaller diameter (less than 100 nm), obtained by chemical and mechanical treatments, have been widely employed to prepare light management films for improving light properties [26]. Especially, the CNF prepared by formic acid (FA) hydrolysis and the followed vacuum filtration exhibited a light transmittance of 90% and a haze of 70%, and the PCE of OSC with this film increased from 15.41% to 16.17% [8]. 

Most cellulose-based light management films used in OSC or OLED are fabricated from lignocelluloses. Compared with lignocelluloses, tunicate cellulose exhibits higher purity (without hemicellulose and lignin), relatively higher crystallinity, and higher aspect ratio [27,28,29]. These advantages indicate that it has great potential to be used for preparation of light management films with better performance. Surprisingly, although TEMPO-mediated oxidation is more suitable to fabricate cellulose film with high transmittance in comparison with other preparation methods [28], the reported tunicate CNF film prepared by TEMPO-oxidation only exhibited 20% light transmittance [28], and the tunicate cellulose nanocrystals film containing 50% konjac glucomannan only showed 52% light transmittance [30]. Importantly, there is no related report regarding the haze of tunicate cellulose film. Therefore, in order to be used as the transparent substrate of OSC, the tunicate-based cellulose film needs to be further explored to obtain both high optical transmittance and high haze. In addition, it is expected to obtain more robust film using tunicate CNF with a higher aspect ratio compared to wood-based CNF. Mechanical strength of substrate film is also of crucial importance for the good workability and durability of OSC.

Therefore, in this work, all-tunicate cellulose film was prepared with a special design, in which TEMPO-oxidized tunicate CNF served as the matrix and the tunicate MFC facilitated the formation of light scattering center. The prepared tunicate CNF/MFC film exhibited both high light transmittance of 85.0% and high haze of 68.3%. Moreover, the haze of the CNF/MFC composite film could be adjusted by controlling the loading amount of MFC. Moreover, the prepared all-tunicate cellulose film had high tensile properties (168 MPa) and good thermostability. Additionally, the PCE of OSC with this film as a light management substrate increased 6.5% compared to the control. Thus, the tunicate cellulose film shows great application potential in the fields of OSC and OLED. 

## 2. Materials and Methods

### 2.1. Materials

The tunic (*Halocynthia roretzi Drasche*) was obtained from the Yellow Sea near Weihai (Shandong, China). NaOH, KOH, acetic acid, NaBr, and absolute ethanol were purchased from Sinopharm Group Chemical Reagent Co., Ltd., Shanghai, China. TEMPO (98%) and NaClO solution (active chlorine content of 14%) were bought from Shanghai Aladdin Bio-Chem Technology Co., Ltd., Shanghai, China, Poly [(2,6-(4,8-bis(5-(2-ethylhexyl)thiophen-2-yl)-benzo[1,2-b:4,5-b′]dithiophene))-alt-(5,5-(1′,3′-di-2-thienyl-5′,7′-bis(2-ethylhexyl)benzo[1′,2′-c:4′,5′-c′]dithiophene-4,8-dione))] (PBDB-T), and 3,9-bis(2-methylene(3-(1,1-dicyanomethylene)-indanone))-5,5,11,11-tetrakis(4-hexylphenyl)dithieno-[2,3-d:2′,3′-d′]-s-indaceno[1,2-b:5,6-b′]dithiophene (ITIC) were purchased from Solarmer Materials Inc., Beijing, China. An aqueous solution of PEDOT: PSS (CLEVIOS P VP AI 4083) was supplied by Beijing Bailingwei Technology Co., Ltd. PDINO was obtained from Derthon Optoelectronic Materials Science Technology Co., Ltd., Shenzhen, China. Chlorobenzene (CB) and methyl alcohol were supplied by Sigma-Aldrich LCC. (St. Louis, MO, USA). All reagents were used without any further purification.

### 2.2. Purification of Tunicate Cellulose

The purification of tunicate cellulose was carried out following the reported procedure with a slight modification [31,32]. First, the tunic was soaked with 5 wt.% NaOH solution for 12 h to remove the inner membrane and the treated tunic was further cut into pieces (~1 × 1 cm^2^). Subsequently, the lumpy pieces were immersed in 5 wt.% KOH solution for 12 h to remove protein. After washing, the tunic was stirred by a magnetic stirrer in deionized water at 60 °C for another 12 h, during which 5 mL acetic acid and 10 mL NaClO solution were added every 2 h for bleaching. The former two steps for removing protein and bleaching were repeated three times until the solid residue was completely white. Then, the solid residue was washed thoroughly with deionized water to neutral pH. Finally, the purified tunicate cellulose was disintegrated with a blender and kept in a sealed bag at 4 °C for further use.

### 2.3. Fabrication of Tunicate MFC and CNF

The tunicate cellulose suspension with a consistency of 10 g/L was homogenized through a high-pressure homogenization (AH-PILOT, ATS Engineering Ltd., Suzhou, China) at 400 bar for 10 passes, and the microfibrillated cellulose (MFC) suspension was obtained. TEMPO-oxidized tunicate cellulose nanofibrils (CNF) were prepared according to the procedure reported in the previous literature [33,34]. TEMPO (0.032 g) and sodium bromide (0.2 g) were separately added to 200 mL tunicate cellulose suspension (10 g/L) with continuous magnetic stirring. After that, NaClO solution (10 g) was slowly added dropwise to the suspension. This suspension was stirred at room temperature (25 °C) for 8 h, and the pH was maintained in the range of 10–10.5 with 0.5 M NaOH solution. Subsequently, the reaction was terminated with 200 mL absolute alcohol, and the obtained oxidized cellulose was filtered and washed with deionized water to neutral pH. Finally, the tunicate CNF suspension was obtained by homogenizing the oxidized cellulose suspension at 800 bar for 10 passes.

### 2.4. Fabrication of Cellulose Films

The CNF suspension with a consistency of 0.04 wt.% was treated with an ultrasonic dispersion machine (200 W, 40 kHz, Kunshan Ultrasonic Instrument Co., Ltd., Suzhou, China) for 30 min. Then, the CNF suspension was vacuum filtered with a filter membrane (pore size of 0.22 μm), and the obtained wet CNF film was dried at 60 °C in an oven for 12 h. The MFC films were prepared following the same procedure. To prepare cellulose films with different optical properties, different amounts of MFC suspensions (0.01 wt.%) were added to the CNF suspension (0.04 wt.%), and the corresponding MFC dosages were 1 wt.%, 2 wt.%, 3 wt.%, 4 wt.%, and 5 wt.% to the dry weight of CNF, respectively. The preparation and drying of the CNF/MFC composite films followed the same procedure as mentioned above.

### 2.5. Preparation of Organic Solar Cell Devices

As illustrated in Figure 1a, the organic solar cell device was constructed with the tunicate CNF/MFC film (3 wt.% MFC to CNF), ITO (Indium tin oxide), PEDOT: PSS, PBDB-T: ITIC, PDINO, and Al, in a bottom-up order. First, the ITO glass was ultrasonically cleaned twice with deionized water for 10 min each, then cleaned with acetone and isopropanol using the same method. After that, the ITO glass was blow-dried with nitrogen. Then, the polydimethylsiloxane (PDMS) glue was evenly applied to the back of the ITO glass sheet with a spin applicator at 4000 rpm for 30 s. Subsequently, the cellulose film was carefully pasted to the back of the ITO glass. After the glue was cured, the obtained device of solar cell was subjected to UV/ozone treatment for 3 min. Then, the PEDOT: PSS solution was spin-coated onto ITO glass at 3000 rpm for 30 s, and annealed at 150 °C for 15 min. Before use, the PEDOT: PSS solution was filtered with a membrane (pore size of 0.45 μm). Upon completion of the operation, the glass sheet was transferred to a closed nitrogen glove box. The active layer material PBDB-T: ITIC (1:1, total 15 mg/mL) was dissolved in chlorobenzene, and this solution was subsequently spin-coated onto PEDOT: PSS layer at 3200 rpm for 35 s to form a coating with 100 nm thick, then annealed at 130 °C for 10 min. Finally, PDINO was spin-coated above the PBDB-T: ITIC layer at 3000 rpm for 25 s, and an 80 nm thick aluminum electrode was evaporated onto the top of the device.

The conventional device structure of solar cell ITO/PEDOT: PSS/PBDB-T: ITIC/PDINO/Al without the tunicate CNF/MFC film was prepared as the control (Figure 1b), following the same procedure as mentioned above.

### 2.6. Characterization

The morphology of CNF was examined by a high-resolution transmission electron microscope (TEM, H-7600, Hitachi, Tokyo, Japan) at an accelerating voltage of 100 kV. Before the TEM test, the 0.001 wt.% CNF suspension was dropped on a carbon-supported copper grid, then dyed with a 2 wt.% uranyl acetates solution for 2 h. The diameter distributions of MFC and CNF samples were calculated using the Nano Measure Software, respectively. The morphology of MFC and composite film was observed using a cold-field emission scanning electron microscope (SEM, S-4800, Hitachi, Tokyo, Japan). Before the SEM test, the samples were attached to a stage with conductive tape and coated with gold for 90 s. In order to observe the cross-sectional morphology of the sample, the film was embrittled by liquid nitrogen. The light transmittance of tunicate cellulose films was measured using a spectrophotometer with an integrating sphere (U-4100, HITACHI, Tokyo, Japan). The haze was measured using a haze meter (NDH-5000, Nippon Denshoku, Japan) according to the standard of ISO 14762-1999. Fourier transform infrared (FTIR) spectra of cellulose samples were obtained by using a spectrometer (Nicolet 6700, Thermo Fisher, Waltham, MA, USA) in attenuated total reflection (ATR) mode with a wavenumber range from 4000 cm^−1^ to 750 cm^−1^. The X-Ray Diffraction (XRD) patterns of the freeze-dried cellulose, CNF and MFC samples were recorded on an X-ray diffractometer (D8 Discover, Bruker, Germany) with the scattering angle (2θ) range of 5–60°, and the scanning speed of 40 °/min. The crystallinity index (CrI) of cellulose samples was calculated according to the Segal method [35] with the subtraction of background. The tensile strength of cellulose films was measured by a universal testing machine according to the standard of TAPPI T497 with a testing speed of 5 mm/min. Each sample was measured at least three times. The thermal stability of samples was measured by a thermogravimetric analyzer (Q600, TA Instruments, New Castle, DE, USA) at a heating rate of 10 °C/min in nitrogen atmosphere. The temperature ranged from 25 °C to 600 °C. The elements of tunicate raw materials and tunicate cellulose were tested by an elemental analysis instrument (VarioEL III, Elementar, Langenselbold, Germany). The current density and voltage (J-V) curves of the solar cells were measured under standard AM 1.5G sunlight (100 mW/cm^2^). The Keithley 2400 workstation and Newport Oriel class A solar simulator was calibrated with monocrystalline silicon reference cells certificated by National Institute of Metrology (NIM).

## 3. Results and Discussion

High light transmittance and high optical haze are two most important properties of light management films for OSC [8]. Before the preparation of tunicate cellulose films, tunicate cellulose needs to be isolated from tunic. Based on the literature, tunic is mainly composed of cellulose (around 60%), protein (about 40%), and a little amount of lipid and ash [36,37]. Herein, alkali pretreatment was used to remove protein and other impurities, and the elements of raw tunic and the purified tunic (tunicate cellulose) are shown in Table 1. Initially, the protein content in the tunic was estimated at 26.75% based on the content of N element. After pretreatment, the protein content was largely decreased to 0.88%, indicating that most of the protein was removed, and the main component of the purified tunic was cellulose. As shown in Figure 2a, the obtained tunicate cellulose is white and slice-like. SEM image (Figure 2b) clearly shows its microstructure of the bonded single fibers with a diameter of about 0.4 μm. Furthermore, the MFC and CNF were prepared from the tunicate cellulose by mechanical homogenization and TEMPO-mediated oxidation plus homogenization, respectively. According to the SEM and TEM images, the average diameter of MFC and CNF samples are 0.38 ± 0.15 μm and 22.50 ± 2.36 nm, respectively (Figure 2c–f). In addition, the digital image of 0.5 wt.% CNF suspension is more transparent than that of MFC suspension with the same consistency. This phenomenon was because CNF, with a much smaller diameter, has much less light scattering compared to MFC [26], and more visible light could pass through the CNF suspension. In contrast, the size of MFC is much larger than the wavelength of light, which is conducive to scattering light.

Moreover, chemical structures of tunicate cellulose, MFC and CNF samples were investigated by FTIR. The same characteristic peaks at 3333 cm^−1^ and 2901 cm^−1^ in all spectra were attributed to −OH stretching and symmetric stretching of C−H, respectively (Figure 2g) [28,30,34]. The peaks at 1161 cm^−1^ and 1111 cm^−1^ were caused by the asymmetric C−O−C bridge stretching and anhydroglucose asymmetric ring stretching, respectively [28,30,34]. Moreover, the peaks at 1059 cm^−1^ and 705 cm^−1^ were attributed to the skeletal vibration of C−O−C pyranose rings and the out of plane bending for Iβ in cellulose, respectively [34]. Compared with tunicate cellulose and MFC, there is a new peak at 1604 cm^−1^ in the spectrum of CNF, which corresponds to a COO− vibration [30,34], confirming the introduction of carboxyl groups on cellulose chains by TEMPO-mediated oxidation.

In addition, the crystalline structures of tunicate cellulose, MFC and CNF were tested by XRD analysis. In all patterns (Figure 2h), the peaks at 14.76° and 16.76° are assigned to (1ī0) and (110) planes, respectively [28,30,34]. The peak around 22.96° is assigned to (200) plane of cellulose Ι structure. According to the XRD pattern, the calculated crystallinity index (CrI) values of tunicate cellulose, MFC and CNF are 85.1%, 85.4%, and 78.4%, respectively. The results show that tunicate cellulose has a high crystallinity (85.1%), and interestingly, homogenization did not have negative impact on the crystallinity of MFC (85.4%). The relatively lower CrI (78.4%) of CNF may be related to the excessive degradation of crystalline regions of cellulose during TEMPO-oxidation, and this phenomenon is consistent with previous reports [38].

All-tunicate cellulose films were prepared by adding a small amount (1–5 wt.%) of MFC to CNF, to tune the optical properties of the obtained CNF/MFC films. Figure 3a shows that when the tunicates were covered by the films, the pure CNF film is highly transparent, while the transparency of the composite CNF/MFC films decreased with the increase in MFC dosage, and the pure MFC film exhibited a poor transparence. To quickly see the haze of the cellulose films with naked eyes, the distance between cellulose films and tunicates was kept in 6 cm. As can be seen from Figure 3b, the pure CNF film showed a low haze and the haze of the composite film increased with the increase in MFC dosage from 1 wt.% to 5 wt.%. Meanwhile, the MFC films still displayed an opaque appearance, which was due to the sufficient light scattering [26,39].

To further quantitatively evaluate the optical properties of the tunicate cellulose films, the light transmittance and haze values were measured accordingly. As can be seen from Figure 3c, the light transmittance at 550 nm (the average wavenumber of visible light) of tunicate cellulose films clearly decreased from 90.3% to 79.2% with the increasing dosage of MFC from 0 wt.% to 5 wt.%. As required, the light transmittance of the high transparent substrate for organic solar cells should be higher than 80% [40,41]. Therefore, the MFC dosage should be lower than 5 wt.%. Moreover, shown in Figure 3d are the haze values of the tunicate cellulose films. As can be seen, the haze of the films increased from 39.2% to 68.3% as the MFC dosage increased from 0 wt.% to 3 wt.%, while the haze slightly went up to 71.3% when MFC dosage increased from 3 wt.% to 5 wt.%. The results indicated that the optical properties of the tunicate cellulose film could be adjusted by mixing CNF with MFC, and the addition of small amount of MFC in CNF could increase the light scattering of the composite films. To balance light transmission and haze, the tunicate cellulose composite film with 3 wt.% MFC is more suitable to be used as light management film with both high light transmittance of 85% and high haze of 68.3%.

To further investigate the change mechanism of optical properties, the surface and cross-section structure of the tunicate cellulose films were analyzed. As can be seen from Figure 4a,b, the surface of MFC film was uneven, which was mainly caused by the large size and the irregularity of MFC. The cross-section of the MFC film exhibited many rough micron fibers and micron-sized pores (Figure 4c). It is well documented that the optical properties of cellulose films have an important relationship with the diameter of fibers and pores [42]. Unlike CNF, the diameter of MFC fibers was larger than the wavelength of light, thus most of the incident light could scatter in all directions, showing an almost opaque appearance [39,43]. In addition, the pores in MFC film contributed to light scattering [44], rather than light absorption, thus leading to the low transparency of the MFC film. By contrast, the CNF film had a uniform surface, dense cross-section morphology, and homogeneous nano-sized fibers (Figure 4d–f). Since the diameter of CNF was much smaller than the light wavelength, most of light could pass through the CNF film [6,45], leading to a high light transmittance up to 90.1% and a low haze of 39.2%. Compared with MFC and CNF films, the composite film with 3 wt.% MFC showed the uniformly dispersed MFC inside the matrix of the CNF film (Figure 4g–i). The CNF matrix facilitated the light transmission, and micron-nano-sized pores facilitated the light scattering. This unique structure is described by a schematic diagram of the films in Figure 4j. Therefore, this all-tunicate cellulose composite film with both high light transmittance and high haze was formed, in which CNF acted as a matrix and a small amount of MFC served as the light scattering centers.

Moreover, the mechanical properties of cellulose films are of great importance for their practical application. Figure 5a shows the stress–strain curves of the CNF film, the MFC film, and the all-tunicate cellulose composite film with different dosages of MFC. The pure CNF film exhibited a high tensile strength of 228 MPa, and the tensile strength of the composite films showed a downward trend with the increase in MFC dosage (Figure 5b). In addition, the toughness of the films also exhibited the same trend (Figure 5c). It is believed that the high mechanical strength of the CNF film was mainly due to the nano-sized effect and extensive hydrogen bonds of CNF fibers [25]. Yet, the addition of MFC in CNF film reduced this effect and increased the defects, e.g., micro pores, thus reducing the tensile strength and toughness of the composite film. Especially, the tunicate cellulose composite film with 3 wt.% MFC showed a tensile strength of 168 MPa and toughness of 5.7 MJ/m^3^, and it could still pull up 500 g weights (Figure 5d). The mechanical properties of this composite film were higher when compared to the reported light management cellulose-based films (e.g., cellulose nanopaper prepared based on TEMPO oxidation (the reported tensile strength was 105 MPa and the corresponding toughness was 1.88 J/m^3^), carboxymethyl cellulose/lignocellulosic fiber film (with a tensile strength of 140 MPa and a toughness of 8.51 MJ/m^3^)) [25,43]. Thus, this tunicate cellulose composite film with 3 wt.% MFC also met the strength requirement of light management film for solar cells.

In addition, thermal stability is important for thin films used in the field of electronic devices. Especially, most solar cell electronic devices work at temperatures higher than 150 °C. According to the TG and DTG curves (Figure 6), MFC film exhibited a higher initial decomposition temperature (329 °C) than that of CNF films (306 °C), and the temperature at the maximum weight loss rate (374 °C) was also higher than that of CNF film (341 °C). This result was mainly because the existing carboxyl groups of CNF could cause decarboxylation reactions in the glucuronic anhydride units [28]. For the composite film with 3 wt.% MFC, the thermal degradation temperature (354 °C) lied in between MFC film and CNF film, which was much higher than the required temperature of OSC. This result indicated that the prepared whole tunicate cellulose film could meet the thermal stability requirement of light management films for OSC [28].

Based on the unique light management properties, high mechanical properties and good thermal stability, the tunicate CNF film with 3 wt.% MFC is expected to be used as a light management film in OSC. To verify the effectiveness of this film, the conventional OSC and the OSC with the composite film were prepared accordingly [8,46]. Figure 7a shows the digital images of Glass/ITO, and Glass/ITO with the whole tunicate cellulose composite film, the prepared conventional OSC and the OSC with the composite film. The composite film could be evenly and tightly adhered to the ITO glass by PDMS. Results indicated that the conventional PBDB-T: ITIC-based solar cells exhibited an average power conversion efficiency (PCE) of 10.71%. According to the literature, the PCE of conventional PBDB-T: ITIC-based solar cells was usually higher than 10.5% [46]. Surprisingly, compared with the conventional OSC, the PCE of the cells with the tunicate cellulose film was 11.41%, which was 6.5% higher compared to the control (Table 2). Additionally, the efficiency of a 50 min independent cell was evenly distributed (Figure 7c). In addition, the short-current density (Jsc) of the cell with the composite film was higher than the control (Figure 7b), indicating that more light was utilized. We measured the extracellular quantum efficiency spectrum (EQE) of these two types of cells (Figure 7d) [8]. Both PBDB-T: ITIC standard cells and the cells with the tunicate cellulose films exhibited a wide EQE response with a wave-length range of 300–800 nm. Between 350 nm and 700 nm, the EQE of OSC with the tunicate cellulose film was clearly higher compared to the control, indicating that the OSC with the tunicate cellulose film effectively improved the mobility of the device, thereby collecting more charge and generating more short-circuit currents. Additionally, the Jsc value obtained by integrating in EQE deviated less than 5% from the data of Jsc in the J-V curve, indicating that the actual Jsc was close to the theoretical calculated value. Therefore, it can be concluded that the prepared all-tunicate cellulose composite film has great potential to be used as a light management film for OSC.

## 4. Conclusions

In summary, this study reported a whole tunicate cellulose film with both high transmittance (85.0%) and high haze (68.3%) at 550 nm, and the film was specially designed with the CNF as the matrix and the MFC as the light scattering center. The optical properties of the composite CNF/MFC film could be easily controlled by adjusting the dosage of MFC. Results indicated that the optical haze of the CNF/MFC film significantly increased from 52.4% to 71.3% with the increase in MFC dosage from 1 wt.% to 5 wt.%. Moreover, this composite film exhibited high tensile strength of 168 MPa, high toughness of 5.7 MJ/m^3^, and good thermal stability (the maximum thermal degradation temperature of 354 °C). In addition, the composite CNF/MFC film was used as light management film in organic solar cells, and its average power conversion efficiency increased 6.5% compared to the control. Although the transmittance and haze of this CNF/MFC film need to be further improved, e.g., more accurately adjust diameters of tunicate CNF and MFC, this work provides a simple strategy to enhance light management of the transparent substrate of organic solar cells for improving power conversion efficiency. Additionally, the prepared whole tunicate cellulose film could expand the application of tunicate cellulose in optoelectronic devices, which has great potential to gradually replace petroleum-based plastics to relieve current energy and environmental pressures of human society.

## Figures and Tables

**Figure 1 nanomaterials-13-01221-f001:**
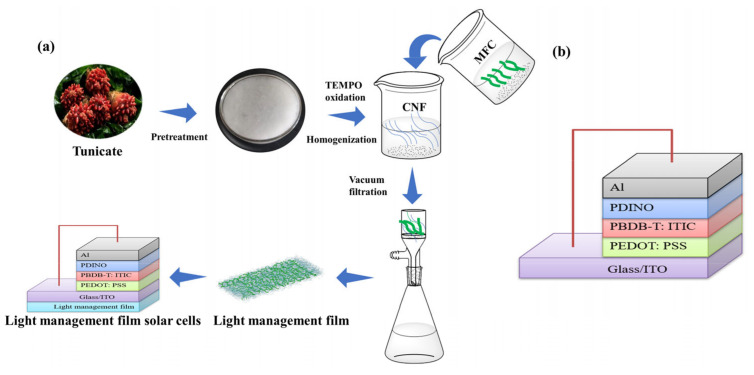
(**a**) Flow diagram for the preparation of tunicate cellulose film and the solar cell with cellulose film. (**b**) Schematic structure of a conventional organic solar cell.

**Figure 2 nanomaterials-13-01221-f002:**
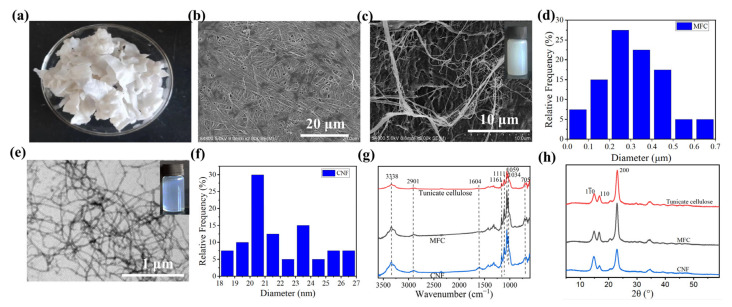
Digital image (**a**) and SEM image (**b**) of tunicate cellulose; SEM image (**c**) and diameter distribution (**d**) of MFC, digital image in the upper right corner is the 0.5 wt.% MFC suspension; TEM image (**e**) and diameter distribution (**f**) of CNF, digital image in the upper right corner is the 0.5 wt.% CNF suspension; FTIR spectra (**g**) and XRD patterns (**h**) of tunicate cellulose, MFC and CNF samples.

**Figure 3 nanomaterials-13-01221-f003:**
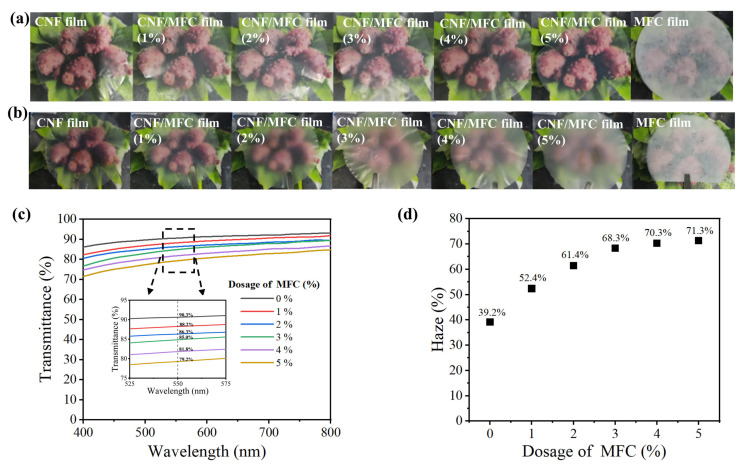
Digital images of CNF film, composite films with different dosage of MFC (1–5 wt.%) and MFC film (the tunicates are covered by the films) (**a**); Digital images of cellulose films (the distance between cellulose films and tunicates is 6 cm (**b**); Light transmittance (**c**) and haze (**d**) of cellulose films with different dosage of MFC.

**Figure 4 nanomaterials-13-01221-f004:**
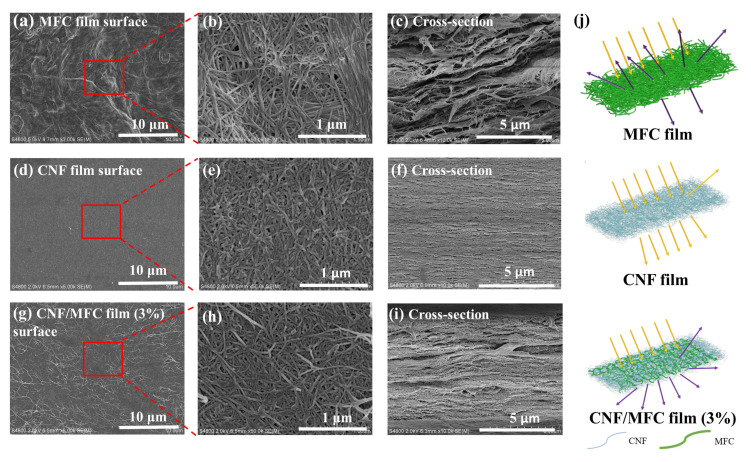
SEM images of MFC film (**a**–**c**), CNF film (**d**–**f**), and CNF/MFC composite film with 3 wt.% MFC (**g**–**i**), Schematic diagrams of MFC film, CNF film and the CNF/MFC composite film (**j**).

**Figure 5 nanomaterials-13-01221-f005:**
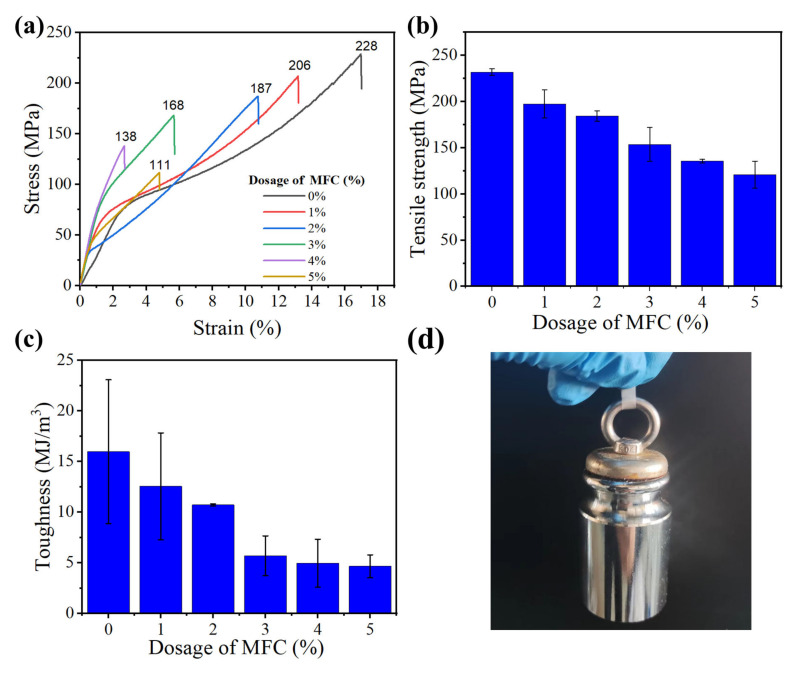
Stress–strain curves (**a**), tensile strength (**b**), and toughness (**c**) of films. (**d**) The composite film (3 wt.% MFC) can pull up 500 g weights.

**Figure 6 nanomaterials-13-01221-f006:**
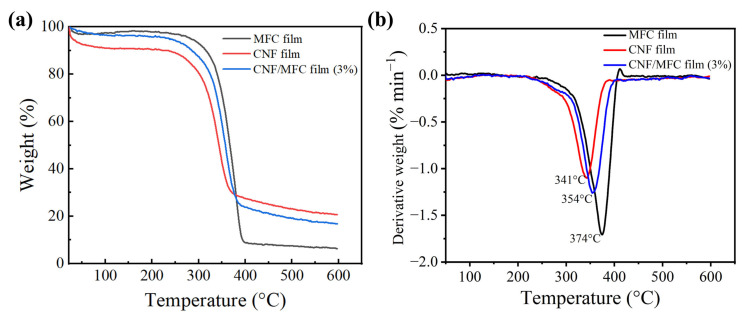
TG (**a**) and DTG (**b**) curves of tunicate cellulose films.

**Figure 7 nanomaterials-13-01221-f007:**
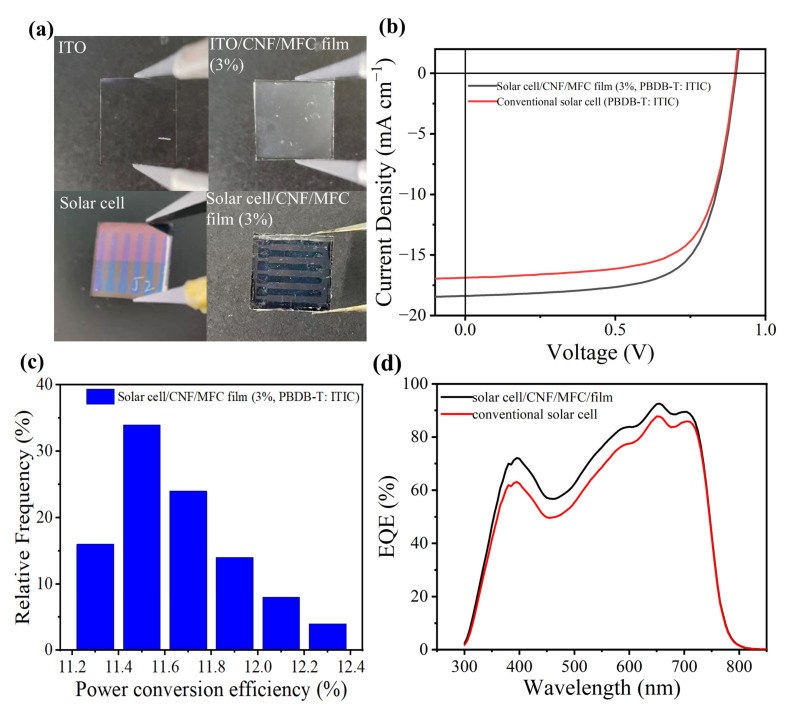
Application of light management thin films in organic solar cells. (**a**) Digital images of Glass/ITO, Glass/ITO/cellulose composite film with 3 wt.% MFC, conventional organic solar cells, and organic solar cells with composite film with 3 wt.% MFC; (**b**) Current density and voltage curves of solar cells; (**c**) Efficiency distribution of 50 optical management film devices; (**d**) The external quantum efficiency (EQE) curves conventional solar cell and solar cell with CNF/MFC film.

**Table 1 nanomaterials-13-01221-t001:** Elemental analyses of tunic and tunicate cellulose.

Samples	N (%)	C (%)	H (%)	S (%)	Protein (%)
Tunic	4.28	37.48	4.698	2.814	26.75
Tunicate cellulose	0.14	40.34	4.588	1.141	0.88

**Table 2 nanomaterials-13-01221-t002:** Photovoltaic parameters of the PBDB-T:ITIC-based conventional solar cells and solar cells with tunicate cellulose composite film.

Samples	Open-Circuit Voltage (Voc) [V]	Short-Circuit Current Density (Jsc) [mA cm^−2^]	Fill Factor (FF) [%]	Power Conversion Efficiency (PCE) [%]
Solar cell	0.901 ± 0.002	17.001 ± 0.250	69.3 ± 1.0	10.71 ± 0.124
Solar cell with film	0.901 ± 0.002	18.375 ± 0.492	68.9 ± 1.3	11.41 ± 0.254

## Data Availability

The data are available from the authors on request.

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
