# Peer review of "All-Tunicate Cellulose Film with Good Light Management Properties for High-Efficiency Organic Solar Cells"

_nanomaterials, 2023, doi:10.3390/nano13071221_

Round 1

Reviewer 1 Report

The authors report the fabrication of a cellulose-based film and its use in organic solar cell devices. Cellulose is an attractive platform for fabricating optoelectronic devices. The manuscript can be of interest for the readers of nanomaterials. However, some issues should be addressed:

-The state of the art discussed in the introduction section is slightly undercut. Please elaborate on this, and  clarify which is the novelty of this study compared to those already existent.

-How many devices of the same type have been prepared?  

-In the conclusion section, please clarify which are the points of strength and the weaknesses of this study. Also, which are the suggestions to guide in designing new cellulose-based components, based on the obtained results?

Author Response

Comments:

The authors report the fabrication of a cellulose-based film and its use in organic solar cell devices. Cellulose is an attractive platform for fabricating optoelectronic devices. The manuscript can be of interest for the readers of nanomaterials. However, some issues should be addressed:

Responses:

We appreciated this reviewer’s positive comments. We have made careful revision according to the reviewer’s valuable comments and suggestions. All revised parts were marked with revision traces in our revised manuscript. The revision was attached, and the point-by-point responses are given below.

Point 1:

-The state of the art discussed in the introduction section is slightly undercut. Please elaborate on this, and clarify which is the novelty of this study compared to those already existent.

Responses:

As suggested, the introduction section was carefully revised and the novelty of this study was further highlighted. The revised section was shown below.

“Cellulose is the most abundant natural polymer material on earth, and it exists in plant cell walls, algae and some tunicate animals. In recent years, cellulose films have been used as light management substrates in solar cells [8,19-22] and OLED [5,23,24], due to their advantages of renewability, biodegradability, good thermal stability, low density and good flexibility. For instance, cellulose film with a light transmittance of 96% and a haze of 60% was obtained using the 2, 2, 6, 6-tetramethylpiperidine-1-oxyl (TEMPO)-mediated oxidation of primary wood fibers and the followed vacuum filtration of the oxidized cellulose fibers. The prepared cellulose film was used as the transparent substrate of organic solar cells (OSC) achieving 10% improvement of power conversion efficiency (PCE) [25]. Except for the primary wood cellulose, the wood-based microfibrillated cellulose (MFC) and cellulose nanofibers (CNF) with smaller diameter (less than 100 nm), obtained by chemical and mechanical treatments, could also be employed to prepare cellulose light management films for improving the light properties [26]. Especially, the CNF prepared by formic acid (FA) hydrolysis and the followed vacuum filtration exhibited a light transmittance of 90% and a haze of 70%, and the PCE of OSC with this film increased from 15.41% to 16.17% [8].

However, the reported cellulose light management films used in OSC or OLED are fabricated with lignocellulose. Compared with lignocellulose, tunicate cellulose exhibits higher purity (without hemicellulose and lignin), relatively higher crystallinity, and higher aspect ratio [27-29]. These advantages indicate that it has a great potential to be used for preparation of cellulose light management film with high performance. Surprisingly, although TEMPO-mediated oxidation is more suitable to fabricate cellulose film with high transmittance in comparison with other preparation methods [28], the reported tunicate CNF film prepared by TEMPO-oxidation only exhibited 20% light transmittance [28], and the tunicate cellulose nanocrystals film containing 50% konjac glucomannan only showed 52% light transmittance [30]. Importantly, there is no related result regarding the haze of tunicate cellulose film. Therefore, in order to be used as the transparent substrate of OSC, the tunicate-based cellulose film needs to be further explored to obtain both high optical transmittance and high haze. In addition, it is expected to obtain more robust film using tunicate CNF with a higher aspect ratio compared to wood-based CNF. Mechanical strength of substrate film is also of crucial importance for the good workability and durability of OSC.

Therefore, in this work, all-tunicate cellulose film was prepared with a special design using TEMPO-oxidized CNF as the matrix and the tunicate MFC as the former of light scattering center. The prepared tunicate CNF/MFC film exhibited both high light transmittance of 85.0% and high haze of 68.3%), and the haze of the tunicate cellulose film could also be adjusted by the amount of MFC based on the dry weight of CNF. Moreover, the prepared all-tunicate cellulose film had high tensile properties (168 MPa) and good thermostability. Additionally, the PCE of OSC with this film as a light management substrate increased 6.5% compared to the control. Thus, the tunicate cellulose film shows a good application promising in the fields of OSC and OLED devices.”

Point 2:

-How many devices of the same type have been prepared?  

Responses:

We prepared 50 organic solar cells (OSCs) using the all-tunicate cellulose film with light transmittance of 85.0% and haze of 68.3%, and the power conversion efficiency (PCE) of these cells were evaluated. Figure 7c shows the efficiency distribution of the PCE, and the average PCE is 11.41% (Table 2).

Point 3:

-In the conclusion section, please clarify which are the points of strength and the weaknesses of this study. Also, which are the suggestions to guide in designing new cellulose-based components, based on the obtained results?

Responses:

Thank you for pointing this out. As suggested, the strength and weakness of this work were clarified, and the corresponding suggestions were given in the revised manuscript.

The revised conclusion is shown below.

“In summary, this study reported a whole tunicate cellulose film with both high transmittance (85.0%) and high haze (68.3%) at 550 nm, and the films was specially designed with the CNF as the matrix and the MFC as the light scattering center. The optical properties of the composite CNF/MFC film were related to the dosage of MFC, and the optical haze of the film significantly increased with the increase of MFC dosage from 1 wt.% to 5 wt.%, but the optical transmittance of the film was slightly reduced. Moreover, this tunicate cellulose composite film exhibited high tensile strength of 168 MPa, high toughness of 5.7 MJ/m3, and good thermal stability (the maximum thermal degradation temperature of 354 °C). In addition, the tunicate cellulose composite film was used as light management film in organic solar cells, and its average power conversion efficiency increased 6.5% compared to the control. Although the transmittance and haze of this tunicate cellulose film need to be further improved and this can be done by more accurately adjusting diameters of tunicate CNF and MFC in future work, this work provides a simple strategy to enhance light management of the transparent substrate of organic solar cells for improving power conversion efficiency. Also, the prepared whole tunicate cellulose film could expand the application of tunicate cellulose in optoelectronic devices, and has great potential to gradually replace petroleum-based plastics to relieve current energy and environmental pressures of human society.”

Reviewer 2 Report

General: In this work, “Mechanical Robust Tunicate cellulose haze films for efficiency with organic solar cells substrate modification” was investigated. First, carefully correct the English by a native speaker. Accordingly, I strongly recommend that you should rewrite and resubmit this manuscript

Specific Comments

(1)    Abstract

The abstract doesn’t have to result. Can you check to the abstract?   

(2)    Introduction

1.       The objectives and originality of this work were not clearly indicated.

2.       Why did you do something that was not done in the previous paper? (A clear purpose is needed.)

3.       What is the biggest difference from existing technology?

(3)    Results and Discussion

1.       More than the paper purpose and data need. The correct purpose is required.

2.       It is impossible to confirm that the all references are published by the conference.

It should be corrected.

3.       There is no interpretation of the experimental results. (Only simple results are shown)

What do you want to explain with the difference purposes? The thesis will need to be corrected for many words. Overall, the author's thesis was written as an explanation without the back grounds. The references to older papers need to be changed. I don't understand to your paper. I can't check to the references.

So,

Overall, the careful writing is required.

Author Response

Comments:

General: In this work, “Mechanical Robust Tunicate cellulose haze films for efficiency with organic solar cells substrate modification” was investigated. First, carefully correct the English by a native speaker. Accordingly, I strongly recommend that you should rewrite and resubmit this manuscript.

Responses:

We very appreciated this reviewer’s deep insight and valuable suggestions.

As suggested, the English writing has been carefully corrected and revised by a native speaker. All the revised parts were marked with the revision traces in our revised manuscript (Please find the attached file). Also, the point-by-point responses to detailed comments are given below.

Specific Comments

Point (1)    Abstract

The abstract doesn’t have to result. Can you check to the abstract?   

Responses:

As suggested, the abstract has been carefully revised, and more specific data has been supplemented.

The revised abstract is given below.

“Tunicate nanocellulose with its unique properties (such as excellent mechanical strength, high crystallinity, and good biodegradability) has potential to be used for the preparation of light management film with high transmittance and haze. However, the reported tunicate cellulose film only exhibits low transmittance, and there is no report on the haze. Herein, we prepared a whole tunicate cellulose film with tunable haze levels, by mixing tunicate micro-fibrillated cellulose (MFC) and tunicate cellulose nanofibrils (CNF). Then, the obtained whole tunicate cellulose film with updated light management was used to modified the organic solar cell (OSC) substrate, aiming to improve the light utilization efficiency of OSC. Results showed that the dosage of MFC based on the weight of CNF was an important factor to adjust the haze and light transmittance of the prepared cellulose film. When the dosage of MFC was 3 wt.%, the haze of the tunicate cellulose film increased 68.3% compared to the pure CNF film (39.2%). Moreover, the optimum tunicate cellulose film exhibited excellent mechanical properties (tensile strength of 168 MPa, toughness of 5.7 MJ/m3) and high thermal stability, which are also of great importance for the improvement of workability and durability of OSC. More interestingly, the prepared whole tunicate cellulose film with a high haze (68.3%) and high light transmittance (85.0%) was used as an additional layer adhered to the glass substrate of OSC, leading to 6.5% improvement of the power conversion efficiency compared to the control (without the tunicate cellulose film). With the use of biodegradable tunicate cellulose, this work provides a simple strategy to enhance light management of the transparent substrate of OSC for improving power conversion efficiency.”

Point (2)    Introduction

  1. The objectives and originality of this work were not clearly indicated.

Responses:

Thank you for pointing this out. The objectives and originality of this work were further highlighted in our revised manuscript.

Objective: The main purpose of this work is to prepare all-tunicate cellulose film with good light management properties, specifically with both high light transmittance and high haze, and apply this film in organic solar cells (OSC) to improve the power conversion efficiency (PCE).

Originality: The reported tunicate cellulose film in the previous literature only has low light transmittance, and there is no research on the haze properties of tunicate cellulose film. Therefore, the originality of this work lies in the successful preparation of tunicate cellulose film with both high transmittance and high haze by adding a small amount of MFC in CNF film, and the optical properties of the obtained tunicate cellulose film can be adjusted by the dosage of MFC. Finally, the application performances of this film were evaluated in OSC, and the PCE of OSC increased 6.5%, indicating this whole tunicate cellulose film can be used as the light management substrate for OSC.

  1. Why did you do something that was not done in the previous paper? (A clear purpose is needed.)

Responses:

So far, the reported cellulose light management films used in OSC or OLED are fabricated with lignocellulose. Compared with lignocellulose, tunicate cellulose exhibits higher purity (without hemicellulose and lignin), relatively higher crystallinity, and higher aspect ratio [27-29]. These advantages indicate that it has a great potential to be used for preparation of cellulose light management film with high performance. Surprisingly, although TEMPO-mediated oxidation is more suitable to fabricate cellulose film with high transmittance in comparison with other preparation methods [28], the reported tunicate CNF film prepared by TEMPO-oxidation only exhibited 20% light transmittance [28], and the tunicate cellulose nanocrystals film containing 50% konjac glucomannan only showed 52% light transmittance [30]. Importantly, there is no related result regarding the haze of tunicate cellulose film. Therefore, in order to be used as the transparent substrate of OSC, the tunicate-based cellulose film needs to be further explored to obtain both high optical transmittance and high haze.

  1. What is the biggest difference from existing technology?

Responses:

(1) In the existing literature, cellulose light management films are mainly prepared from lignocellulose. In this paper, light management films for OSCs are successfully prepared with tunicate CNF and MFC.

(2) In the existing literature, the light transmittance of tunicate cellulose films is low, and there is no information about haze. In this paper, all-tunicate cellulose film with both high light transmittance (85.0%) and high haze (68.3%), is successfully fabricated using CNF and MFC.

Point (3)    Results and Discussion

  1. More than the paper purpose and data need. The correct purpose is required.

Responses:

In this work, all-tunicate cellulose film was prepared with a special design using TEMPO-oxidized CNF as the matrix and the tunicate MFC as the light scattering center. The obtained whole tunicate cellulose film with updated light management was used to modified the organic solar cell (OSC) substrate, aiming to improve the light utilization efficiency of OSC.

More data on the evaluation of OSC with the tunicate cellulose film was added (Figure 7). The added description is given below.

“In addition, the short-current density (Jsc) of the cell with the composite film was higher than the control (Figure 7b), indicating that more light was utilized. We meas-ured the extracellular quantum efficiency spectrum (EQE) of these two types of cells (Figure 7d). It can be seen that both PBDB-T: ITIC standard cells and the cells with the tunicate cellulose films exhibited a wide EQE response with a wave-length range of 300-800 nm. Between 350 and 700nm, the EQE of OSC with the tunicate cellulose film was clearly higher compared to the control, indicating that the OSC with the tunicate cellulose film effectively improved the mobility of the device, thereby collecting more charge. This was conducive to increasing the short-circuit current. Also, the Jsc value obtained by integrating in EQE deviated less than 5% from the data of Jsc in the J-V curve, indicating that the actual Jsc was very close to the theoretical calculated value.”

  1. It is impossible to confirm that the all references are published by the conference.

It should be corrected.

Responses:

As suggested, we double-checked and revised the references in the revision.

  1. There is no interpretation of the experimental results. (Only simple results are shown)

What do you want to explain with the difference purposes? The thesis will need to be corrected for many words. Overall, the author's thesis was written as an explanation without the back grounds. The references to older papers need to be changed. I don't understand to your paper. I can't check to the references.

So, Overall, the careful writing is required.

Responses:

We appreciated this reviewer’s valuable comments. As suggested, the purpose of this article has been clarified in the introduction section, and the discussion according to the results has also been supplemented in the revised manuscript. Moreover, the references have been updated, and the English writing has been carefully double corrected and revised by a native speaker. All the revised parts were marked with the revision traces in our revised manuscript.

Round 2

Reviewer 1 Report

The authors sufficiently addressed the comments raised by this reviewer. 

Author Response

Comments:

The authors sufficiently addressed the comments raised by this reviewer. 

Responses:

We appreciated this reviewer's valuable inputs, which indeed helped a lot for improving the quality of our manuscript.

Reviewer 2 Report

General: In this work, “Mechanical Robust Tunicate cellulose haze films for efficiency with organic solar cells substrate modification” was investigated. First, carefully correct the English by a native speaker. Accordingly, I strongly recommend that you should rewrite and resubmit this manuscript

Specific Comments

The revised paper provides proper explanations for the questions and comments raised in the previous review. Justification in English grammar is still required, but it can be resolved by the manuscript review service from editorial office. I have no further question or objection for this contribution to be considered as a potential publication in Applied Sciences.

So,

Overall, the careful writing is required.

Author Response

Comments:

General: In this work, “Mechanical Robust Tunicate cellulose haze films for efficiency with organic solar cells substrate modification” was investigated. First, carefully correct the English by a native speaker. Accordingly, I strongly recommend that you should rewrite and resubmit this manuscript

Response:

As suggested, the English of our manuscript has been carefully corrected by a native speaker. Also, we have double checked the English writing of this manuscript.

Specific Comments

The revised paper provides proper explanations for the questions and comments raised in the previous review. Justification in English grammar is still required, but it can be resolved by the manuscript review service from editorial office. I have no further question or objection for this contribution to be considered as a potential publication in Applied Sciences.

So,

Overall, the careful writing is required.

Responses:

We appreciated this reviewer's valuable comments. 

The English writing of our manuscript has been carefully corrected by a native speaker. Also, we have double checked and carefully improved the English writing of this manuscript.